# REF-EMGBENCH:
# BENCHMARKING REFERENCE NORMALIZATION FOR ELECTROMYOGRAPHY DATA

## ABSTRACT

Electromyography (EMG)-based hand gesture recognition is essential for applications in prosthetics, rehabilitation, and human-robot interaction. Despite advances in machine learning, domain shift caused by intersubject variability often leads to degraded model performance when applying trained models to new users. In this study, we revisit the statistical reference normalization methods to mitigate the domain shift in EMG data in a leave-one-subject-out train-test split setting. We systematically benchmark five popular amplitude-based normalization techniques to assess their effectiveness in subject-specific classification with varied datasets and percentages for normalization. Experimental results show that Min-Max and Peak normalization outperform others, yielding higher classification accuracy on EMG data. We further visualize the domain shifts in the feature space throughout the training process and provide an analysis based on EMG signal characteristics. Our findings indicate that proper normalization significantly reduces intersubject variability of EMG samples, enhancing model adaptation and providing insights for bridging domain shifts in future EMG-based gesture recognition research. The benchmark code for domain adaptation approaches on EMG signals is available at `ref-emgbench.github.io`.

## 1 INTRODUCTION

Electromyography (EMG) is a crucial data source to assess muscle activity and predict movement intentions, playing a significant role in physical rehabilitation and the control of prosthetic devices. However, EMG signals often exhibit substantial variability across different subjects and recording sessions. This variability comes from multiple factors, including electrode configuration and placement (Mesin et al. (2009)), perspiration (Abdoli-Eramaki et al. (2012); Winkel & Jørgensen (1991)), temperature fluctuations (Winkel & Jørgensen (1991)), physiological differences (Dellon & Mackinnon (1987); Nourbakhsh & Kukulka (2004), muscle fiber composition (Halaki & Ginn (2012)), blood flow (Halaki & Ginn (2012)), and the amount of tissue between the electrode surface and the muscle (Halaki & Ginn (2012)).

To mitigate variability and extract meaningful and consistent patterns from raw EMG signals, various amplitude normalization methods have been explored since the late 1950s, primarily from a signal processing perspective (Halaki & Ginn (2012)). Common normalization techniques include scaling signals relative to maximal voluntary contractions (Edelstein (1986); Yang & Winter (1983)) or peak values (Allison et al. (1993); Yang & Winter (1984)). These methods aim to standardize signal amplitudes across different subjects and sessions, thereby reducing intersubject and intrasession differences (Halaki & Ginn (2012); Lin et al. (2020)).

However, the challenge of domain shift across subjects in sEMG data remains unresolved, necessitating further investigation into effective preprocessing strategies. Amplitude normalization can play a crucial role in enhancing the generalizability of machine learning models by standardizing signal amplitudes and reducing variability (Kerber et al. (2017); Khushaba (2014); Lin et al. (2020)). However, no previous work has benchmarked the use of different normalization methods in EMG, although normalization methods have shown great promise in dealing with distribution shifts (Du et al. (2017); Ioffe (2015); Li et al. (2018); Côté-Allard et al. (2019)).

In this work, we systematically investigate several EMG data normalization techniques as preprocessing steps and evaluate their impact on the performance of deep learning classifiers in hand gesture recognition tasks based on EMG data. By examining how different normalization methods affect a model's ability to generalize across subjects, we aim to identify the most effective strategies for mitigating domain shift. In this paper, we make the following contributions:

1. We present a comprehensive benchmarking of five statistical normalization methods, evaluating their effectiveness in mitigating intersubject variability for hand gesture recognition based on EMG data.

2. We provide detailed visualizations and analyses that offer insight into the extent to which different normalization methods reduce the distribution shift.

3. We demonstrate that inter-subject reference normalization consistently outperforms intra-subject reference normalization, underscoring the potential of leveraging inter-subject variability as a key contribution to improving EMG-based classification performance.

## 2 REFERENCE NORMALIZATION OF EMG DATA

### 2.1 EMG SIGNALS

EMG signals are captured as multi-channel time-series data, with the number of channels varying significantly. Some systems utilize only a few manually placed electrodes on specific arm muscles (Ozdemir et al. (2022a)), while others can involve up to 128 channels (Geng et al. (2016)). The sampling frequency also varies, typically ranging from a few hundred Hz (Atzori et al. (2014)) to several thousand Hz (Ozdemir et al. (2022a); Yang et al. (2023)). EMG signals, recorded in the microvolt range, are amplified and bandpass filtered to mitigate noise from external sources, such as mechanical interference. These low-voltage, high-frequency signals result from ion movements during neuromuscular excitation (Purves et al. (2001)).

### 2.2 DOMAIN SHIFT IN TIME-SERIES DATA

In time-series data, domain shift refers to the challenge that arises when the statistical properties of the data change between the training and deployment phases, leading to a discrepancy between the training and testing distributions. This shift can significantly degrade model performance, especially in real-world applications where the environment or conditions evolve over time. The domain shift in EMG data usually manifests itself in the form of **concept shift**.

The concept shift (Fan et al. (2024); Zhang et al. (2022)) in EMG data involves changes in the underlying relationship between the EMG features and the output labels over time. This shift can occur due to physiological changes, such as muscle fatigue, or differences in motor control strategies between participants. For instance, a model trained on EMG data from one participant may perform poorly when applied to another participant, or even to the same participant at a later time, due to changes in how muscle signals correspond to the intended gestures. This makes it essential to address concept shift through strategies such as continuous model fine-tuning or the development of algorithms that can adapt to evolving feature-label relationships, ensuring reliable performance over time and across varying conditions.

### 2.3 REFERENCE NORMALIZATION

To address the challenges posed by the domain shift in EMG data, a variety of normalization techniques are employed to standardize the data for more reliable analysis. These methods are often used to reduce noise, smooth signals, or scale the data for consistency across trials or subjects. In addition, a specific normalization process called **reference normalization** adjusts the EMG signal relative to a single individual, which has been shown to have practical improvements in reducing concept shifts across subjects in Lin et al. (2020).

As illustrated in Fig. 3.2, the process of reference normalization begins by computing normalization parameters based on a target dataset, which in our work corresponds to the fine-tuning dataset or data from one of the subjects in the training dataset. Once these parameters are obtained, they are

applied across the entire dataset, transporting the original distribution to the distribution of the target subject. Because EMG signals are recorded as multi-channel time-series data, reference normalization of EMG data is performed on a per-channel basis, using statistical parameters such as the mean, variance, or extremas from channel $i$ for gesture $j$ from subject $k$, to standardize data from the same channel and gesture across a different subject $k'$. This approach contrasts with traditional methods that calculate normalization parameters across the full dataset without accounting for individual variability.

In this work, we apply transfer learning in conjunction with reference normalization using statistical amplitude features. This approach enables deep models to adapt to the specific distribution of the target subject while preserving the generalizability of the pretrained model. The statistical normalization methods used for benchmarking are introduced in the following section.

## 2.4 AMPLITUDE NORMALIZATION METHOD

A wide range of amplitude normalization techniques have been employed to standardize time-series data, each offering unique approaches to managing amplitude variability. In this section, we discuss the most widely used methods: Z-score, Min-Max, Root Mean Square (RMS), Mean Absolute Value (MAV), and Peak normalization. These methods are evaluated in our study to benchmark their performance in gesture recognition tasks using a deep learning model, helping to identify the most effective approach for standardizing EMG data.

### 2.4.1 Z-SCORE

Z-score normalization standardizes EMG signals to have zero mean and unit variance, making it particularly useful for handling data from varying distributions (Koval (2018)). By scaling based on each data point's deviation from the mean relative to the standard deviation, it is less sensitive to outliers compared to methods that rely on the dataset's extrema:

$$\mathbf{Z} = \frac{\mathbf{X} - \mu}{\sigma},$$

where $\mathbf{X}$ is the original signal, $\mu$ is the mean and $\sigma$ is the variance.

### 2.4.2 MIN-MAX

Min-Max normalization scales EMG signals to a fixed range, typically between 0 and 1, which helps standardize signal ranges across subjects. By adjusting data based on its minimum and maximum values, this method can compress smaller magnitudes, potentially reducing the impact of noise. However, it is more sensitive to extreme values, which may distort scaling in the presence of outliers. Despite this, Min-Max normalization remains common in EMG due to the typically stable amplitudes in such signals (Tkach et al. (2010); Lin et al. (2020)):

$$\mathbf{Z} = \frac{\mathbf{X} - \min(\mathbf{X})}{\max(\mathbf{X}) - \min(\boldsymbol{X})}.$$

### 2.4.3 ROOT MEAN SQUARE (RMS)

RMS normalization scales EMG signals based on their root mean square value, providing a meaningful measure of signal energy relevant in both time and frequency domains (Phinyomark et al. (2012)). Although this method is sensitive to outliers due to the squaring of values, it remains a valuable tool for assessing signal magnitude and energy in EMG biosignal analysis:

$$\mathbf{Z} = \frac{\mathbf{X}}{\text{RMS}(\mathbf{X})},$$

$$\text{RMS}(\mathbf{X}) = \sqrt{\frac{1}{N} \sum_{i=1}^{N} \mathbf{X}_i^2}.$$

where $N$ is the number of timesteps used to calculate the RMS value.

### 2.4.4 MEAN ABSOLUTE VALUE (MAV)

MAV normalization, similar to RMS, scales signals by their mean absolute value but reduces sensitivity to outliers by using absolute values instead of squared values. However, this method may be less effective at minimizing the impact of small noise values, as it treats all deviations from zero equally (Phinyomark et al. (2012)):

$$\mathbf{Z} = \frac{\mathbf{X}}{\text{MAV}(\mathbf{X})},$$

$$\text{MAV}(\mathbf{X}) = \frac{1}{N} \sum_{i=1}^{N} |\mathbf{X}_i|.$$

### 2.4.5 PEAK

Peak normalization adjusts EMG signals based on their maximum value, making it particularly effective for applications emphasizing peak amplitudes Allison et al. (1993); Yang & Winter (1984). While this method is sensitive to outliers, such sensitivity can be beneficial in tasks focused on peak performance or heightened neuromuscular activity during gestures:

$$\mathbf{Z} = \frac{\mathbf{X}}{\max(\mathbf{X})}.$$

## 3 EXPERIMENT

### 3.1 DATASET

We evaluated the proposed normalization techniques using three publicly available sEMG datasets: CapgMyo (DBb) and the NinaPro DB3 and DB5 databases. These datasets were chosen for their diversity in subjects, hand gestures, and recording conditions, providing a strong basis for assessing the effectiveness of normalization methods in reducing intersubject variability. Since EMG signals have applications for both individuals with amputations and able-bodied users, we selected datasets that reflect both user groups. Additionally, we included a variety of electrode configurations, ranging from high-density gelled flexible circuit boards to individually placed bipolar electrodes and wearable low-density electrode solutions.

CapgMyo (DBb): Introduced in Geng et al. (2016) and expanded upon in Du et al. (2017), this dataset consists of sEMG recordings from 10 subjects performing 8 distinct gestures, captured using 128 high-density electrodes. The recordings were segmented into 250 ms windows, resulting in 25,600 samples. The acquisition system includes 8 modules, each equipped with 16 electrodes.

NinaPro DB3 and DB5: The NinaPro datasets Atzori et al. (2014); Pizzolato et al. (2017) are widely used in sEMG research, particularly for prosthetics and human-computer interaction. DB3 contains recordings from 11 subjects with transradial amputations, using 12 bipolar Delsys Trigno electrodes. DB5 includes data from 10 able-bodied subjects, recorded with two Myo Armbands (each with 8 bipolar electrodes around the forearm). After windowing, DB3 contains 64,426 samples, while DB5 has 39,597 samples.

### 3.2 EXPERIMENT SETTING

As shown in Fig. 3.2, we first split each dataset using a leave-one-subject-out approach, designating the left-out individual as the target or reference subject. The first 1%, 5%, or 10% of the target subject's data (stratified by gesture) is used as a fine-tuning dataset. The remaining data from the target subject is split into the first 50% for the validation set and the second 50% for test.

For each amplitude normalization method discussed in Sec. 2.4.1, we compute the normalization parameters, such as mean, standard deviation, minimum, and maximum, based on the fine-tuning set prior to converting the EMG signals into heatmap images. Once these parameters are determined, the entire dataset, including the pre-training, fine-tuning, testing, and validation sets, is normalized to generate new subsets and subsequently converted to heatmaps as the inputs to the model. The selected deep learning model, ResNet18, is initially trained on the pre-training dataset, followed

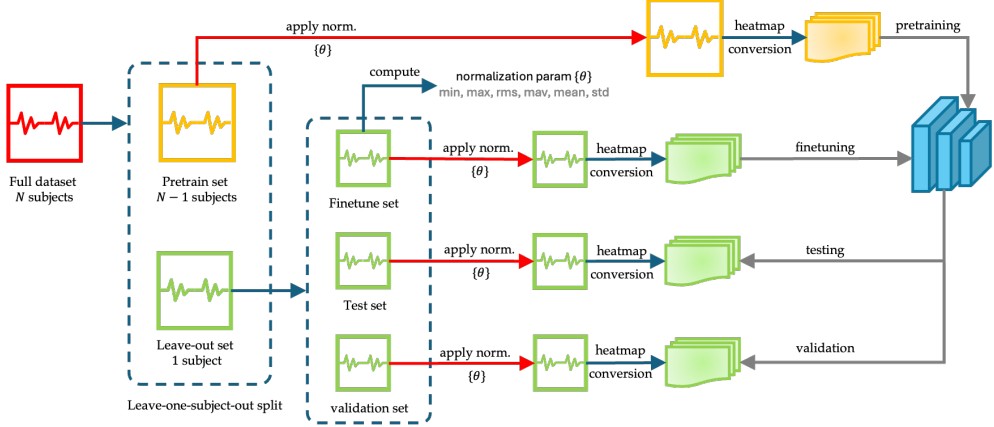

Figure 1: Flow chart for benchmarking process

by fine-tuning on the fine-tuning set, with periodic validation to save the best-performing model. Testing is conducted after completing the fine-tuning process.

To benchmark the model in a controlled setting, we report the classification and domain shift metrics at the end of the training process, rather than selecting the best-performing model. Specifically, the model is trained for 50 epochs on the pre-training set and 300 epochs on the fine-tuning set. Optimization is performed using the Adam optimizer with a learning rate of $1e-5$ and a constant learning rate scheduler.

## 3.3 EVALUATION METRICS

To quantitatively evaluate the effectiveness of normalization methods, we use the following metrics: accuracy, Area Under the Receiver Operating Characteristic Curve (AUROC), Maximum Mean Discrepancy (MMD) and Kullback-Leibler (KL) divergence.

Test accuracy refers to the proportion of correctly classified instances among all test instances, providing a straightforward measure of the model's predictive performance. While accuracy provides a straightforward measure of the model's ability to correctly classify instances, it does not account for potential imbalances in the data or the distribution shifts between training and test domains.

AUROC assesses the model's ability to distinguish between classes across all classification thresholds. This metric provides insight into the trade-off between true positive and false positive rates. In our multiclass setting, AUROC is computed using the one-vs-rest (OvR) approach, where we calculate the AUROC for each class treated against all other classes, and average the results.

MMD is a statistical measure used to quantify the difference between the probability distributions of the source (training) and target (testing) domains. A lower MMD value indicates a smaller domain shift, suggesting that the normalization method effectively aligns the feature distributions.

KL divergence measures how one probability distribution diverges from a reference distribution. It is used to quantify the discrepancy between the feature distributions of the source and target domains, with lower values indicating better alignment.

## 3.4 VISUALIZATION OF DOMAIN SHIFT

To gain qualitative insights into how normalization affects the distribution of sEMG features across subjects, we employed the t-Distributed Stochastic Neighbor Embedding (t-SNE) and the Wasserstein distance matrix for visualization of the domain shift.

t-SNE is a dimensionality reduction method that projects high-dimensional data into a two-dimensional space while preserving the local structure of the data. By visualizing the feature representations using t-SNE, we can observe the clustering patterns of different subjects before and after normalization, providing visual evidence of a reduced domain shift.

The Wasserstein distance, also known as the Earth Mover's distance, is a measure of the distance between two probability distributions. We computed the Wasserstein distance matrix for the feature distributions of all gesture classes to quantify category-wise domain shifts. Visualizing this matrix helps to understand the effectiveness of normalization methods in aligning the feature spaces across different subjects.

Detailed information on visualization results can be found in Fig. A.2, Fig. A.3 and Sec. 3.5.2.

## 3.5 RESULTS AND ANALYSIS

### 3.5.1 ADAPTABILITY TO TARGET SUBJECT

The comparison of the five amplitude normalization methods, evaluated using classification and domain shift metrics, is presented in Table 1 and Table 2. The results indicate that Min-Max and Peak normalization consistently outperform the other three methods by a significant margin. Both demonstrate superior performance in terms of higher test accuracy and AUROC scores, as well as lower KL divergence and MMD across all three datasets.

To further illustrate the impact of normalization methods and the proportion of fine-tuning and reference normalization data, we provide parallel coordinate plots for each dataset in Fig. A.1. These plots offer a clear visual interpretation of how different normalization techniques influence the evaluation metrics. It can be observed from the plots that Min-Max normalization consistently results in the lightest color across all three datasets, indicating stronger performance, with Peak normalization closely following. The visual representation underscores the robustness of these two methods in reducing domain shift while maintaining high classification accuracy.

Across all normalization methods, an increase in the percentage of fine-tuning data leads to improvements in both classification accuracy and domain shift metrics. This trend is particularly evident with Min-Max and Peak normalization, where more fine-tuning data (10%) result in stronger performance metrics across datasets.

| Dataset | % for RN & FT | Z-score | Min-Max | RMS | MAV | Peak |
|---|---|---|---|---|---|---|
| CapgMyo DBb | 1 | 87.77 / 98.09 | **99.87 / 100.00** | 88.07 / 98.14 | 86.46 / 97.68 | 94.84 / 99.62 |
| | 5 | 89.98 / 98.56 | 99.65 / 100.00 | 90.12 / 98.55 | 89.16 / 98.42 | **96.32 / 99.66** |
| | 10 | **91.56 / 98.84** | 99.65 / 100.00 | **91.70 / 98.88** | 90.83 / 98.91 | 95.66 / 99.64 |
| | w/o RN & FT | | | 39.95 / 79.56 | | |
| NinaPro DB3 | 1 | 24.98 / 65.88 | 42.45 / 78.24 | 24.52 / 65.72 | 23.86 / 64.83 | 25.20 / 67.51 |
| | 5 | 31.85 / 71.50 | 53.11 / 83.28 | 31.66 / 71.26 | 31.33 / 70.91 | 35.92 / 74.18 |
| | 10 | **36.27 / 74.96** | 60.76 / 85.32 | **36.59 / 75.12** | 36.59 / 75.16 | **39.69 / 76.85** |
| | w/o RN & FT | | | 10.59 / 50.80 | | |
| NinaPro DB5 | 1 | **46.30 / 82.59** | 71.48 / 94.16 | **46.16 / 82.47** | 44.48 / 81.36 | **61.52 / 90.49** |
| | 5 | 35.27 / 75.28 | 66.98 / 92.84 | 35.05 / 75.22 | 33.82 / 74.26 | 48.56 / 84.27 |
| | 10 | 37.03 / 76.56 | 66.51 / 92.35 | 36.75 / 76.53 | 35.68 / 75.33 | 49.48 / 85.50 |
| | w/o RN & FT | | | 29.91 / 70.21 | | |

Table 1: Test accuracy(↑) / AUROC(↑) comparison across 3 datasets

### 3.5.2 DOMAIN ADAPTATION VISUALIZATION

We plot the t-SNE graph over the pre-training and fine-tuning process to visualize how the deep classifier progresses with the five normalization methods in Fig. A.2. Compared to other normalization methods, Min-Max consistently leads to better class separation, as evidenced by the distinct clusters that emerge as early as pre-train epoch 1 and improve progressively throughout the fine-tuning process. Peak normalization also leads to improved clustering; by the end of fine-tuning (epoch 300), some methods like Peak begin to show improved separability, though they do not reach the clarity observed with Min-Max normalization. Z-score, RMS, and MAV normalization exhibit less distinct clustering in the earlier epochs, with more overlapping points between classes.

We plot the categorical Wasserstein distance matrix in Fig. A.3 by comparing the distribution difference between the prediction of the model for the training along with the test set and the one-hot

| Dataset | % for RN & FT | Z-score | Min-Max | RMS | MAV | Peak |
|---|---|---|---|---|---|---|
| CapgMyo DBb | 1 | 4.77 / 27.06 | 0.17 / **0.32** | 4.64 / 26.99 | 5.18 / 30.39 | 1.93 / 14.33 |
| | 5 | 3.80 / 22.34 | **0.13** / 0.35 | **3.80** / 22.21 | 4.03 / 24.29 | 1.64 / 10.30 |
| | 10 | **3.10 / 18.86** | 0.16 / **0.32** | 3.90 / **18.94** | **3.13 / 21.18** | **1.51 / 9.87** |
| | w/o RN & FT | | | 35.47 / 42.29 | | |
| NinaPro DB3 | 1 | 55.08 / 68.72 | 36.35 / 58.96 | 55.79 / 67.87 | 56.83 / 68.53 | 53.04 / 67.74 |
| | 5 | 50.26 / 58.27 | **29.16** / 40.29 | 50.48 / 58.89 | 50.82 / 59.47 | 47.27 / 55.43 |
| | 10 | **45.48 / 53.94** | 30.55 / **31.96** | 45.02 / 54.96 | 45.43 / 54.69 | **42.72 / 53.51** |
| | w/o RN & FT | | | 10.35 / 57.34 | | |
| NinaPro DB5 | 1 | **34.25 / 46.68** | **16.55 / 24.12** | 34.52 / 46.55 | 36.05 / 47.58 | **22.24 / 34.45** |
| | 5 | 42.79 / 57.53 | 19.39 / 27.56 | 42.75 / 57.48 | 44.03 / 58.26 | 30.80 / 47.34 |
| | 10 | 40.31 / 57.51 | 20.68 / 28.01 | 39.95 / 58.01 | 41.67 / 59.46 | 28.45 / 47.60 |
| | w/o RN & FT | | | 74.74 / 38.96 | | |

Table 2: KL-Divergence($1e^{-1}$, ↓) / MMD($1e^{-3}$, ↓) comparison across 3 datasets

label distribution, categorized by gestures to visualize class-wise distributional changes. The category labels and distance range can be found in Fig. A.3.

Across the different epochs, Min-Max normalization exhibits the most distinct diagonal pattern, indicating better alignment between the predicted and actual label distributions. This suggests that Min-Max normalization is effective in reducing the distributional gap between training and testing sets, leading to more accurate predictions.

Compared to Min-Max and Peak normalization, methods such as Z-score, RMS, and MAV show less clear diagonal patterns in earlier epochs, especially during pretraining. This suggests slower convergence and less effective alignment between the training and testing distributions, leading to poorer performance in the earlier stages of the training process. This difference may be attributed to the reliance on mean values or standard deviations in Z-score, RMS, and MAV normalization, as opposed to the use of minimums and maximums in Min-Max and Peak normalization. The latter methods may facilitate faster convergence and more effective distribution alignment, particularly for EMG signals, which often exhibit distinct magnitude differences between channels activated or not activated by neuromuscular signals (Yang et al. (2023)).

### 3.5.3 INTER- AND INTRA-SUBJECT NORMALIZATION

To assess the impact of reference subject selection on classification performance, we compared test accuracy and AUROC between two settings: intra-subject normalization, where normalization parameters are computed using data from the left-out subject, and inter-subject normalization, where parameters are calculated using data from a randomly selected subject in the pre-training set. The results are presented in Table 3.

Min-Max and Peak normalization demonstrate superior performance across all settings, achieving near-perfect AUROC scores(100.00) in several cases. These methods consistently outperform Z-score, RMS, and MAV normalization, especially in higher fine-tuning percentages, indicating their robustness in handling both intersubject and intrasubject normalization scenarios.

Notably, inter-subject normalization (RN subj. ≠ FT subj.) generally produces better results than intra-subject normalization (RN subj. = FT subj.) across all normalization methods and data splits. A possible explanation for this is that using data from a different subject to compute the normalization parameters introduces additional variability into the data distribution, enhancing the model's ability to generalize. In contrast, restricting normalization to the left-out subject might limit this variability, resulting in slightly reduced generalization performance. This finding highlights the potential benefit of leveraging inter-subject variability for improved classifier generalization.

| % for RN & FT | RN subj. & FT subj. | Z-score | Min-Max | RMS | MAV | Peak |
|---|---|---|---|---|---|---|
| 1% | RN subj. = FT subj. | 87.77 / 98.09 | 99.87 / 100.00 | 88.07 / 98.14 | 86.46 / 97.68 | 94.84 / 99.62 |
| | RN subj. ≠ FT subj. | 89.67 / 98.59 | 99.96 / 100.00 | 89.92 / 98.54 | 89.46 / 98.53 | 97.34 / 99.92 |
| 5% | RN subj. = FT subj. | 89.98 / 98.56 | 99.65 / 100.00 | 90.12 / 98.55 | 89.16 / 98.42 | 96.32 / 99.66 |
| | RN subj. ≠ FT subj. | 94.79 / 99.60 | **99.97 / 100.00** | 95.02 / 99.62 | 93.55 / 99.45 | 97.81 / 99.91 |
| 10% | RN subj. = FT subj. | 91.56 / 98.84 | 99.62 / 100.00 | 91.70 / 98.88 | 90.83 / 98.91 | 95.66 / 99.64 |
| | RN subj. ≠ FT subj. | **95.89 / 99.76** | 99.91 / 100.00 | **96.06 / 99.77** | **94.60 / 99.62** | **98.28 / 99.94** |
| w/o RN & FT | | 39.95 / 79.56 | | | | |

Table 3: Test accuracy(↑) / AUROC(↑) comparison on inter- and intra-subject normalization with CapgMyo DBb

# 4 RELATED WORK

## 4.1 EMG SIGNALS AND DISTRIBUTION SHIFT

EMG signals, although non-invasive and recorded from the skin, are subject to various forms of non-stationarity, which introduces significant challenges in generalization across datasets. These non-stationarities arise from biological variability and sensor placement, resulting in what is known as **distribution shift** (Campbell et al. (2024)). Distribution shifts refer to changes in the statistical properties of EMG signals between training and testing phases, complicating machine learning models' ability to generalize. Specifically, a difficult type of distribution shift for machine learning algorithms to deal with is concept shift, where the probability of an output $y$ is different given the same $x$. Common causes include variations in muscle location (Dellon & Mackinnon (1987); Nourbakhsh & Kukulka (2004)), electrode placement (Mesin et al. (2009)), and skin properties like impedance (Rask-Andersen et al. (2019)). Reference normalization techniques aim to reduce concept shift by adjusting data distributions to be more consistent between domains, often improving model robustness.

## 4.2 HAND GESTURE RECOGNITION WITH EMG DATA

Extensive research has been dedicated to training machine learning models for EMG-based gesture recognition, leveraging both publicly available datasets (Lu et al. (2022); Islam et al. (2024); Wei et al. (2021); Hye et al. (2023)) and novel datasets collected by researchers (Côté-Allard et al. (2019); Yang et al. (2023); Ozdemir et al. (2022b); Li et al. (2023); Wang et al. (2023); Zhang et al. (2023); Xu et al. (2023); Algüner & Ergezer (2023); Sussillo et al. (2024)). Classification work for the benchmarking of feature extraction methods and dimension reduction methods have been performed for sEMG signals, including mean absolute value, root mean square, Wilson amplitude, zero-crossing rate, wavelength, power spectrum analysis, short-time Fourier transform, and wavelet decompositions (Phinyomark et al. (2012); Ozdemir et al. (2020)).

While many studies report results based on randomized train-test split accuracy (Hye et al. (2023); Algüner & Ergezer (2023); Sri-Iesaranusorn et al. (2021)) and k-fold cross-validation (Zhang et al. (2022); Ozdemir et al. (2022b); Fatimah et al. (2021); He & Jiang (2020); Kim et al. (2019)), where data from the training, validation, and test sets are randomly sampled from the same subjects, such methods may not adequately reflect real-world scenarios. In practice, it is often preferable for the validation and test sets to consist of data collected either after the training data from the same subject (referred to as train-test splits for time series, or TSTS), or from entirely different subjects excluded from the training set, as evaluated by leave-one-subject-out cross-validation (LOSO-CV). These data splitting strategies offer more robust assessments of model performance by introducing out-of-distribution generalization challenges.

In the case of TSTS, testing with data collected after the training set introduces potential distribution shifts caused by factors such as variations in gesture execution, fatigue (Liu et al. (2021); Chua et al. (2024)), perspiration (Abdoli-Eramaki et al. (2012)), electrode displacement (de Talhouet & Webster (1996)), drying or changes in ionic concentrations of hydrogel or electrolyte gels (Sousa et al. (2023)), and changes in electrode adherence to the skin (Chi et al. (2010)). Similarly, LOSO-CV introduces variability arising from inter-individual differences in body size, muscle morphology (Dellon & Mackinnon (1987)), and variations in skin impedance and adipose tissue distribution (Rask-

Andersen et al. (2019)). Studies employing randomized or mixed data splits, where evaluation data may precede training data, risk reporting inflated accuracies that do not reflect real-world generalization capabilities in practical EMG classifier deployments. In our work, we benchmark using only TSTS and LOSO-CV, which are highly useful metrics for real-world applications.

### 4.3 REFERENCE NORMALIZATION OF EMG DATA

To address distribution shifts, several normalization techniques have been proposed to reduce intersubject variability. Kerber et al. (2017) introduced a peak-based normalization method, which performed well for simple gestures but struggled as the number of gestures increased. Similarly, Khushaba (2014) developed a canonical correlation analysis (CCA)-based framework, achieving 82.96% accuracy for 12 finger movements by extracting style-independent features.

Building on these efforts, Lin et al. (2020) proposed a min-max normalization approach for intersubject EMG-based hand gesture recognition. This method recalibrates training data using the minimum and maximum amplitudes from a new user's signals, effectively reducing domain shift. With a convolutional neural network (ConvNet) and leave-one-subject-out cross-validation (LOSO-CV), the approach achieved 85.09%, 88.97%, and 94.53% accuracy across datasets, outperforming standard normalization techniques and rivaling state-of-the-art transfer learning methods like progressive neural networks and adaptive batch normalization (Côté-Allard et al. (2019)). Unlike transfer learning, which requires more data from the target domain, this normalization method generalizes with minimal data, making it more suitable for many real-world applications.

Normalization techniques like the one proposed by Lin et al. (2020). provide an effective solution to inter-subject variability, addressing one of the primary challenges in EMG data classification. By reducing the domain shift between users, these methods enable robust generalization across diverse populations without sacrificing real-time performance, further advancing the applicability of EMG-based gesture recognition in practical scenarios. However, the lack of benchmarking on multiple datasets and evaluations on the decreases in distribution shift reduces the potential impact of the work.

## 5 CONCLUSION

In this study, we systematically benchmark five amplitude-based normalization methods to address the domain shift challenge in EMG-based hand gesture recognition. Our findings highlight the significant role of normalization in improving model generalization across subjects, with Min-Max and Peak normalization methods demonstrating superior performance in adapting under intersubject variability and enhancing classification accuracy.

Through visualizations and analyses of feature space evolution during training, we showed how effective normalization mitigates domain shifts, facilitating better adaptation of machine learning models to new users. Experiment results show that inter-subject normalization consistently outperformed intra-subject normalization, emphasizing the value of leveraging inter-subject variability. These insights contribute to advancing the development of robust, generalizable models for EMG-based applications in prosthetics, rehabilitation, and human-robot interaction in the future.

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

# A APPENDIX

## A.1 PARALLEL COORDINATE GRAPH

## A.2 T-SNE VISUALIZATION

## A.3 WASSERSTEIN DISTANCE MATRIX

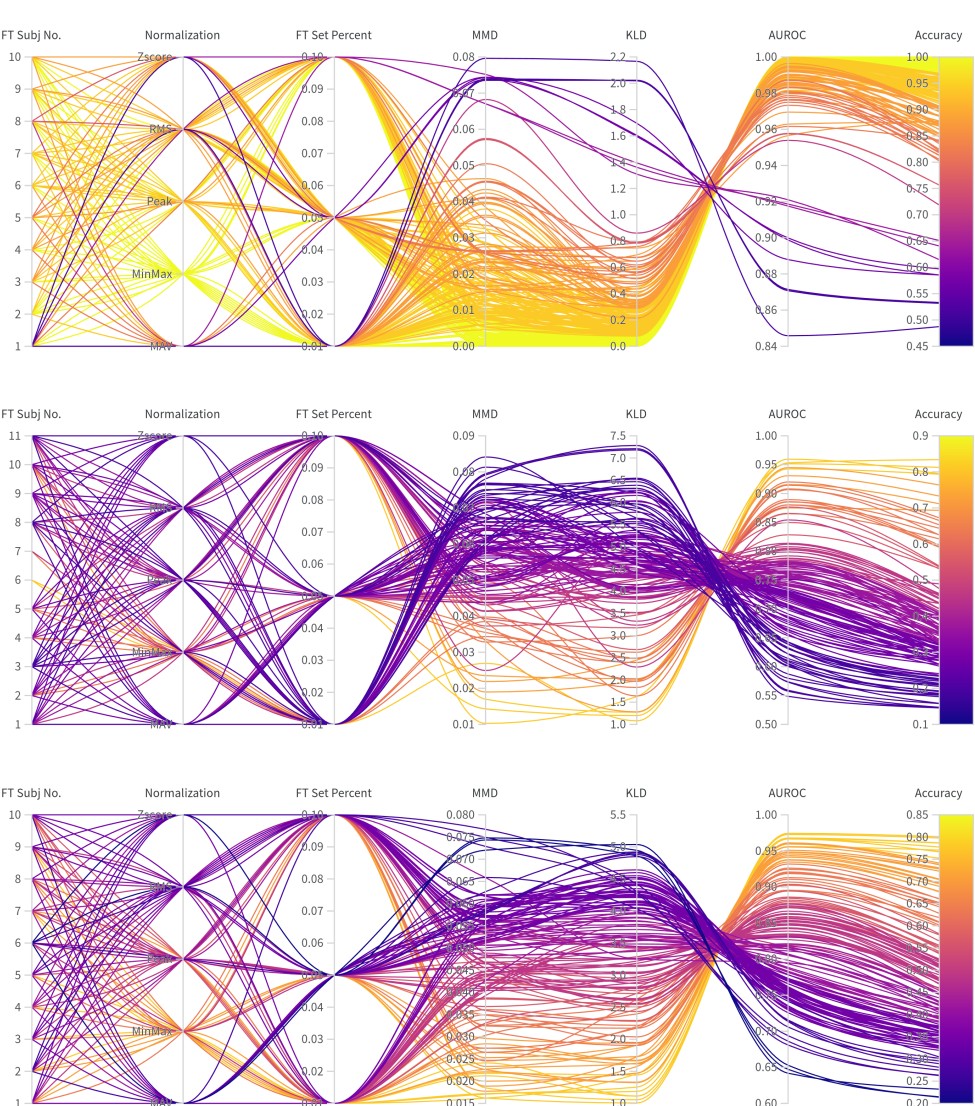

Figure 2: The parallel coordinate graphs on 3 datasets. Top - CapgMyo DBb, middle - NinaPro DB3, bottom - NinaPro DB5. The first column represents the subject number used for testing, finetuning and normalziation; the second column shows the normalization methods; the third column represents the percentage of dataset used for finetuning and normalization; the fourth to the last columns are the evaluation metrics.

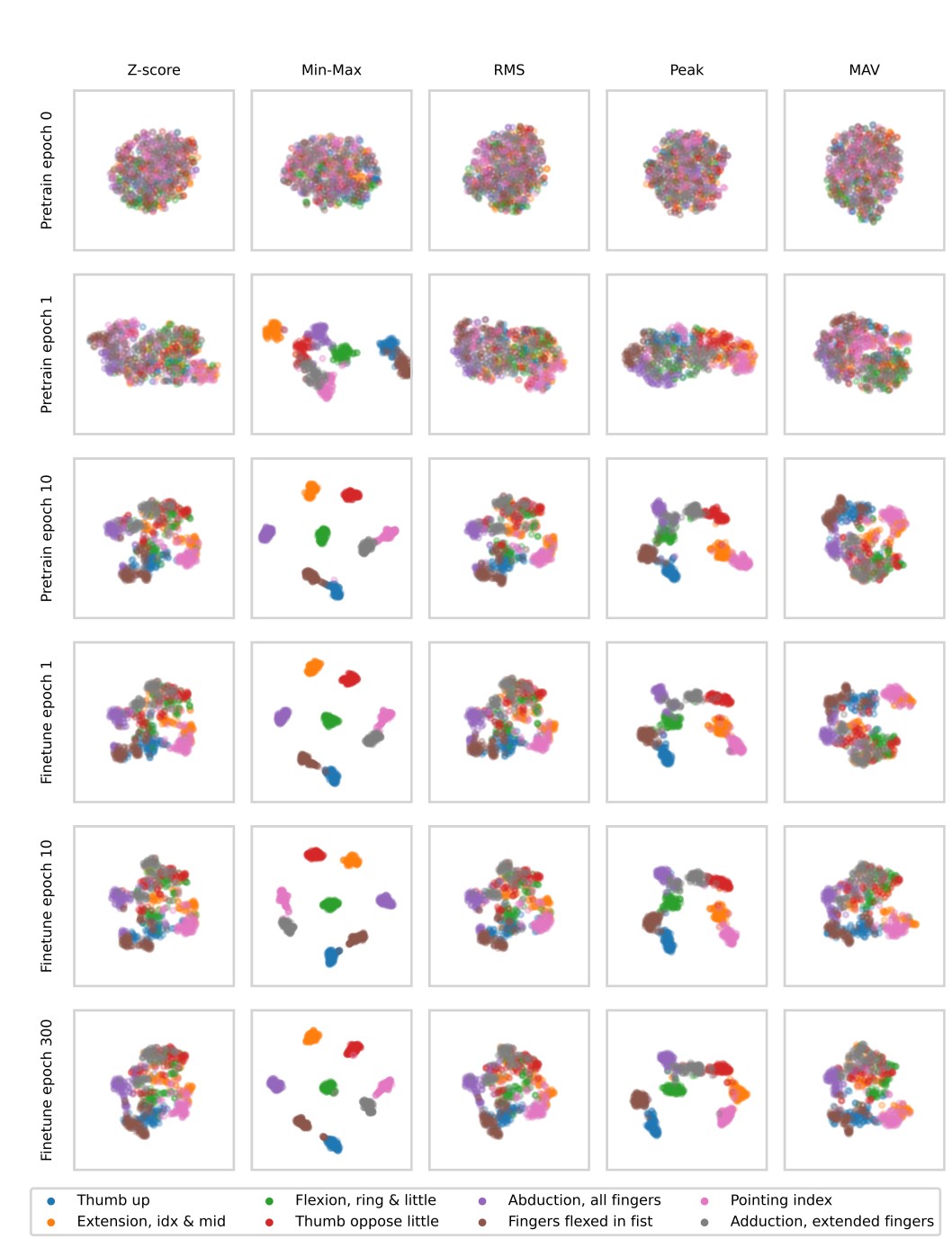

Figure 3: t-SNE evolution during pretraining and finetuning with CapgMyo dataset

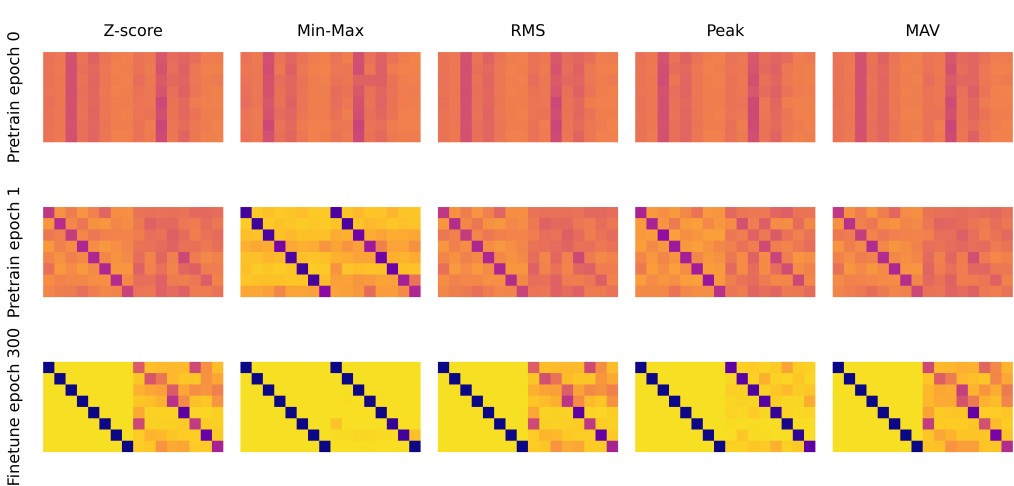

Figure 4: Categorical Wasserstein distance matrix at the begining and end of training with CapgMyo dataset.

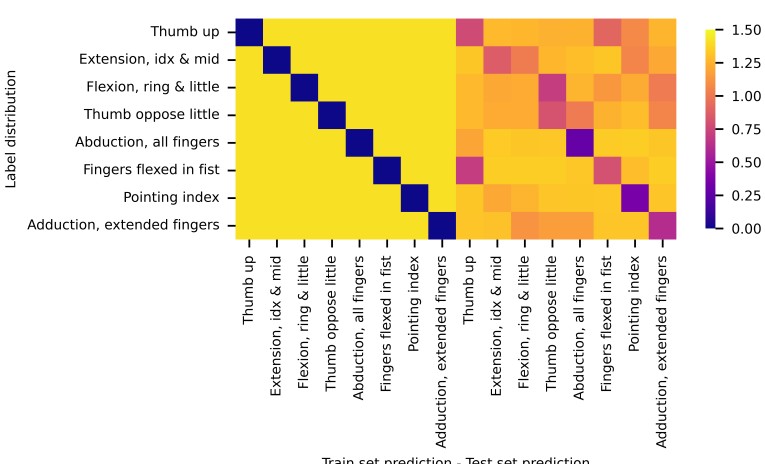

Figure 5: Categorical Wasserstein distance matrix formulation

