# OpenReview forum: "Ref-EMGBench: Benchmarking Reference Normalization for Electromyography Data"
_ICLR.cc/2025/Conference — ICLR 2025 Conference Withdrawn Submission_

### Official Review · Reviewer_zjie · 2024-11-02

**Soundness:** 2
**Presentation:** 1
**Contribution:** 1
**Rating:** 1
**Confidence:** 5

**Summary:**

This manuscript compares five popular amplitude-based normalization techniques using ResNet-18 with a fine-tuning strategy in a leave-one-subject-out setting. The experiments were performed using three publicly available EMG-based gesture datasets. The authors aim to address the domain shift challenge existing among different human participants.

**Strengths:**

There are three major findings: (1) The pretraining-finetuning strategy helps to address the challenge, (2) Inter-subject normalization is better than intra-subject normalization, and (3) Min-max normalization works best.

**Weaknesses:**

1. Comparison Fairness: In the 'Experiment Setting' section (lines 207-210), the authors mention that 1%, 5%, or 10% of the target subject's data is used for fine-tuning, with the remaining data split between validation and testing. However, selecting different percentages of samples from left-out participants would yield different test datasets, which may compromise the fairness of the comparison. Additionally, it's unclear if experiments were conducted with different random seeds to ensure robustness. Most importantly, for revision, a very clear presentation on train-validation-test split along with pre-training & fine-tuning strategies, and intra/inter-subject normalization techniques for different scenarios is required.

2. Limited Baseline Models: It seems that the paper only includes ResNet-18 as a baseline model, which limits the comparison to a single architecture.

3. Incomplete Code and Documentation: In the code repository, all the contents in a few folders, for example, 'diffusion_augmentation' and 'Model' seem unrelated to this study, such as strategies for text-to-image stable diffusion. Please correct me if I am wrong. This inclusion creates confusion from the main focus. For instance, it's unclear if the models in the 'Model' folder are used in this work, which needs further clarification.

4. Method Comparison: The authors do not compare their method with other existing normalization techniques (e.g., [1]), which do not require heavy pre-training and fine-tuning.


[1] Lin, Yuzhou, et al. "A normalisation approach improves the performance of inter-subject sEMG-based hand gesture recognition with a ConvNet." 2020 42nd annual international conference of the IEEE engineering in medicine & biology society (EMBC). IEEE, 2020.

**Questions:**

Some of questions overlap with the section 'Weakness':

1. Could the authors clarify the train-validation-test split strategy when training 'w/o RN & FT' in Table 1? Also, in Table 1, what 'RN' stands for?

2. How are EMG signals converted to heatmaps, and is the conversion code available in the repository? More detailed descriptions are required.

3. Have additional baseline models other than ResNet-18 been considered for the comparative analysis?

4. In the repo shared with this manuscript, is the 'diffusion_augmentation' folder relevant to this paper’s experiments? Similarly, are all models in the 'Model' folder used in this work?

---

### Official Review · Reviewer_zZe8 · 2024-11-03

**Soundness:** 3
**Presentation:** 2
**Contribution:** 3
**Rating:** 6
**Confidence:** 5

**Summary:**

This paper addresses the challenge of domain shift in EMG-based hand gesture recognition due to intersubject variability. The authors benchmark five amplitude-based normalization methods—Z-score, Min-Max, RMS, MAV, and Peak—to evaluate their effectiveness in improving model generalization. Using a leave-one-subject-out approach with ResNet18, normalization parameters from a fine-tuning set are applied across all data subsets. Results show that Min-Max and Peak normalization consistently perform best, enhancing classification accuracy. Visual analyses illustrate that effective normalization can help distinguish different classes. The study highlights the advantage of inter-subject normalization over intra-subject normalization for better generalization.

**Strengths:**

The paper addresses the critical issue of domain shift in EMG-based gesture recognition, emphasizing the importance of normalization due to signal variability across subjects and sessions. It demonstrates originality by benchmarking five amplitude-based normalization methods and highlighting the value of inter-subject normalization, providing new insights into enhancing model adaptability.
The significance of Min-Max and Peak normalization were shown to consistently improve model generalization. The evidence supporting inter-subject normalization over intra-subject normalization highlights the benefit of leveraging inter-subject variability to enhance model adaptability. The paper’s introduction is clear and well-structured. Overall, the paper makes a meaningful contribution to advancing EMG signal processing by demonstrating how targeted normalization can enhance model robustness and generalization.

**Weaknesses:**

While the paper provides valuable insights, several areas need improvement to enhance clarity and impact. The methodology section, in particular, lacks detailed explanations and sufficient justification for certain choices. Additionally, some visualizations, such as those with overlapping lines, are not effective and make interpretation difficult. Addressing these issues would greatly improve the overall clarity and comprehensibility of the paper.

**Questions:**

While the paper is generally well-explained, several points could benefit from clarification for a more comprehensive understanding:

1.	In the abstract, the statement, *“Experimental results show that Min-Max and Peak normalization outperform others, yielding higher classification accuracy on EMG data,” would be more impactful with quantitative details. Including specific numbers and comparisons with other methods would clarify how much better these methods performed.
2.	Please clarify the rationale for converting normalized data into heatmap images. Why was this specific representation chosen, and how does it influence the performance of the deep learning model?
3.	The paper mentions periodic validation during fine-tuning to save the best-performing model. Could the authors specify the criteria or metrics used to determine the “best-performing” model during this phase?
4.	How were target subjects chosen for the leave-one-subject-out approach? Was the selection random, or were there specific criteria to ensure consistency in the normalization process?
5.	Min-Max and Peak normalization may be less effective for data from amputees, who have unique EMG signal characteristics like altered muscle control and inconsistent amplitudes. Did the authors consider how these methods perform on amputee data if present in the dataset?
6.	The observation that inter-subject normalization outperforms intra-subject normalization is interesting. Could the authors delve deeper into why added variability from a different subject enhances generalization and helps models adapt to new data?
7.	In Section 3.5.3, could the authors provide more detail on what “intra-subject normalization” entails? Does it refer to using the same individual's data across multiple sessions or repeated movements (limited to a single session)?
8.	Figure 2 contains many overlapping lines, making it difficult to interpret. Simplifying the visualization by averaging results across participants or showcasing one or two examples might improve clarity. Additionally, the colors used are hard to track, which complicates following individual results.
9.	Figures 3 and 4 appear blurry. Overall, the visualizations could be improved for better comprehension.

---

### Official Review · Reviewer_qW8S · 2024-11-04

**Soundness:** 2
**Presentation:** 2
**Contribution:** 2
**Rating:** 3
**Confidence:** 4

**Summary:**

This paper performs a fairly extensive empirical study on five
different normalization methods used in EMG especially to transfer
models to new users.

**Strengths:**

Strengths

- Code is made available

- several datasets are compared and they seem to find consistency in rankings of the normalization methods

- The paper is pretty clearly written though many details are missing (see questions)

**Weaknesses:**

Weaknesses
- Study is only empirical - Would be a stronger paper if the paper could relate the best normalization methods to the various reasons they cite for the problem (e.g. electrode placement, perspiration, temperature changes, ... amount of tissue between muscle and electrode).

- Plots of the Raw EMG from different users and how the normalizations affect the signals would be informative.

- Some details of the methods are missing in the paper (see questions below)

- The figure captions need more details (see questions below)

- There are limited general ML insights and the insights for EMG would be more valuable if related to the kinds of variability in EMG

- Unclear how important the training data from other subjects is (see questions below) (other than being helpful for setting the normalization parameters)

**Questions:**

One way	of normalizing would be, for example using Z-score, to Z-score each training subject using their own mean and variance and then Z-score the test and validation data using the mean and variance from the	FT data, but it	appears that you instead compute mean and variance from the FT data and use that as the mus and sigmas (per channel)  to normalize all the training, FT, and	test and valid data.  Is that correct?

You say "50 epochs on pre-training and then 300 on fine-tuning" -	Are the same learning parameters used for each of how much does pre-training affect results?   What do results look like if you only use the other subjects for computing normalization parameters and then train on the FT data?

Did you just use the validation set for early stopping or for setting other parameters?

In Figure 3, it	appears	that the confusable classes differ between the Min-Max methods and the others (the others seem to	have pink and orange close, while min-max has pink and gray close)  Can	you explain this?

More explanation is needed in the figure captions.  There is not enough information to fully understand what is being shown.  For example, In Fig 3, are the shown data from the pretraining (training subject) data or the finetuning (test subject) data  or from the test or validation data?

---

### Official Review · Reviewer_6LBa · 2024-11-04

**Soundness:** 2
**Presentation:** 3
**Contribution:** 1
**Rating:** 3
**Confidence:** 3

**Summary:**

This study evaluates five amplitude-based normalization techniques to reduce domain shifts in EMG-based hand gesture recognition, finding that Min-Max and Peak normalization effectively enhance classification accuracy.

**Strengths:**

Addresses an important real-world application in prosthetics, rehabilitation, and human-robot interaction.
Provides valuable insights that can significantly inform and improve EMG-based gesture recognition applications.

**Weaknesses:**

Limited novelty for a top-tier machine learning conference; however, the work is highly suited for journals or benchmark tracks due to its thorough evaluation and practical relevance.

**Questions:**

Does the leave-one-subject-out approach align more closely with domain generalization or domain adaptation, given the variability in EMG data across subjects?
Would the reported results hold if the authors considered experimental setups and model selection scenarios as discussed in [1,2]?

[1] Gulrajani, I., & Lopez-Paz, D. (2020). In search of lost domain generalization. arXiv preprint arXiv:2007.01434.
[2] Gagnon-Audet, J. C., Ahuja, K., Darvishi-Bayazi, M. J., Mousavi, P., Dumas, G., & Rish, I. (2022). WOODS: Benchmarks for out-of-distribution generalization in time series. arXiv preprint arXiv:2203.09978.

---

### Note · Authors · 2024-11-16

**Comment:**

We thank all the reviewers for their detailed feedback and comments. We will incorporate the suggestions and continue to improve our work!

**Withdrawal Confirmation:**

I have read and agree with the venue's withdrawal policy on behalf of myself and my co-authors.